# Higher mindfulness and lower self-absorption predict greater compassion within romantic relationships

Theresa Waclawek [1,2] ✉, Georges Monette [3] & Astrid Schuetz[1]

Investigations into mindfulness and well-being have increasingly turned toward social outcomes such as compassion. However, research is needed to understand how mindfulness relates to compassion in real-world contexts. Moreover, the mechanisms underlying this relationship remain insufficiently understood, with self-related processes—particularly self-absorption and interdependent self-construal—representing important candidates. To address these gaps, we conducted a preregistered, ten-day diary study with $n = 230$ adults ($M_{age} = 40$, SD = 12; 50% female, 49% male, and 1% N/A) in heterosexual romantic relationships ($M_{length\ of\ relationship} = 15$ years, SD = 11 years; $n = 114$ dyads with both members contributing). Partner ratings of compassion were collected to capture how mindfulness relates to a person's compassion as perceived by close others in everyday life. We found that distinct facets of mindfulness showed different associations with compassion across levels of analysis: within persons, daily state increases in mindful acceptance corresponded with higher compassion ($B = 0.19$, SE = 0.06, $p = 0.002$, 95% CI[0.07, 0.31]), whereas between persons, greater trait mindful attention predicted greater compassion ($B = 0.52$, SE = 0.17, $p = 0.002$, 95% CI[0.19, 0.85]). We also found support for self-absorption as a mediating mechanism, as greater mindfulness (state and trait) was associated with more compassion through lower self-absorption. Although we expected this indirect effect to be stronger among individuals higher in interdependent self-construal, this moderation was not supported. Together, these findings highlight the importance of distinguishing state and trait processes in understanding the link between mindfulness and compassion in real-world contexts, and identify self-absorption as a key mechanism. Future work can address limitations of this work by investigating its generalizability to other populations.

Mindfulness is often described as "paying attention in a particular way: on purpose, in the present moment, and nonjudgmentally"[1]. The potential for mindfulness to promote well-being has motivated extensive research over the past few decades. Empirical evidence supports associations between mindfulness and a myriad of intrapersonal psychological and physical health outcomes[2,3]. However, attention has more recently turned to the question of how mindfulness relates to interpersonal outcomes. As we are social creatures, the quality of our interpersonal experiences in part defines the quality of our lives[4,5]. Therefore, it is important to understand what conditions allow for social interactions that enhance our well-being.

Converging evidence links mindfulness with prosocial behaviour— both when considering mindfulness as a dispositional tendency and as an outcome of mindfulness-based interventions[6–8]. A prosocial response that is particularly relevant to well-being is compassion. Compassion, or a "sensitivity to suffering in self and others with a commitment to try to alleviate and prevent it"[9], has been increasingly acknowledged as a key ingredient to a healthy society[10,11]. It is particularly important because it is a prosocial response focused on alleviating suffering[12], and it is during these times of suffering when social support is especially vital to well-being[13,14]. Mindfulness has deep historical ties to compassion and has been considered a foundation for compassionate behaviour within its traditional Buddhist context[15,16]. Therefore, understanding the link between mindfulness and compassion is central to understanding how mindfulness may exert interpersonal effects that contribute to well-being. In terms of recent empirical work, mindfulness has been reliably linked with compassion. More specifically, recent meta-analyses have found that mindfulness training increases

[1]Department of Psychology, University of Bamberg, Bamberg, Germany. [2]Department of Psychology, York University, Toronto, ON, Canada. [3]Department of Mathematics and Statistics, York University, Toronto, ON, Canada. ✉e-mail: waclawek@yorku.ca

compassion specifically more so than other prosocial behaviours. For example, a meta-analysis by Berry et al. [8] found that mindfulness training without explicit ethical instruction reliably increased compassionate helping but not instrumental or generous helping. Similarly, a meta-analysis by Kreplin et al. [17] found that mindfulness training increased compassion but did not have other prosocial outcomes such as reduced aggression.

Despite the evidence for a relationship between mindfulness and compassion, it is still not clear how these processes unfold in people's daily lives. First, the majority of mindfulness research thus far has measured mindfulness via trait measurements relying on retrospective self-report [18]. Measuring mindfulness as a trait using retrospective report could be a problem because recall bias may limit ecological validity [19,20]. Additionally, trait mindfulness is understood as indicating how frequently and consistently individuals experience mindful states [21,22]. Mindfulness, however, is often conceptualized as an ongoing process [23], and evidence implies that it fluctuates across time [22,24] and context [22]. While trait measurements are informative, measuring mindfulness also as a state rather than just as an overall tendency allows for a more fine-grained understanding of how mindfulness varies with outcomes of interest. State mindfulness itself has been associated with well-being outcomes [19,24–26], and work examining associations of mindfulness using both state and traditional retrospective trait measurements found that state measurements were associated with larger effect sizes than the traditional retrospective trait measures when predicting well-being outcomes [19,24,25].

The issue of ecological validity, the question of variation over time, as well as the problem of longer-term retrospective bias could be addressed via the use of experience-sampling methods. These methods produce both within-person (state) and between-person (trait) measurements, but increase ecological validity by reducing recall bias, and capture changes in constructs or relationships of interest in real-world settings [20,27]. Daily diary methods have been successfully employed in research on both mindfulness [28–31] and compassion [32–34] (though diary studies on compassion so far have been focused on self-compassion, rather than compassion toward others). Research comparing the use of experience-sampling methods with traditional retrospective self-report for measuring both mindfulness [25] and compassion [35] has shown that experience-sampling methods show greater sensitivity and have been able to detect theorized outcomes when retrospective trait measurements have not. Research is needed to bring these areas together to understand the association between mindfulness and compassion in daily life. Thus, an investigation into mindfulness and compassion as they are associated in people's daily lives and real environments using experience-sampling methods is both timely and much needed.

Second, a further important critique of previous studies is that they have primarily employed self-ratings of prosocial behaviour. These ratings are prone to social desirability bias [36], and indeed, self-ratings of prosocial behaviour from previous mindfulness research have been larger than observer ratings [6]. Further, if mindfulness is also self-reported, then there is the added issue of common source bias [37]. Additionally, beyond methodological concerns, an important question is whether or not mindfulness is associated with compassion in a way that is perceptible to others. That is, if one important function of compassionate behaviour is to provide support in times of need, does mindfulness detectably increase compassionate behaviour as perceived by those in need? Currently, compassion outcomes have been most often studied as compassion towards the self (self-compassion) rather than towards others [38]. To reduce reliance on a single source and to have a better understanding of how mindfulness relates to compassion as it is experienced interpersonally, other-ratings of compassion are needed. Close, important relationships with frequent contact are especially important in dealing with stressors [39], making romantic couples who live together an ideal sample. Although ratings from romantic partners are not free from bias, they provide an important interpersonal perspective on compassion. While there is limited work to date on mindfulness and compassion towards others within romantic relationships, mindfulness has been indirectly positively associated with one's partner's relationship satisfaction through

acceptance [40]. Mindfulness has also been associated with one's own relationship satisfaction via self-reported caring for one's partner [41]. Further, studies employing experience-sampling methods have found daily mindfulness to positively relate to perceived partner responsiveness as rated by the romantic partner [31], and to self-reported empathy within the context of a romantic relationship [42]. Research is needed, however, to understand how mindfulness relates to compassion towards one's partner within the context of a romantic relationship - and how it relates to partner perceptions. Taken together, to address these previous research gaps and to improve generalizability to daily life, we will use experience-sampling methods to investigate how fluctuations in a person's daily mindfulness relate to how compassionate their partner perceives them to be in their usual daily environment. Our first hypothesis is that mindfulness will positively relate to compassion towards one's partner in daily life.

In addition to understanding the relationship between mindfulness and compassion in daily life, there have been calls to better understand the mechanisms underlying this relationship [6,7,43]. Self-related processes have frequently been proposed as key mechanisms underlying the effects of mindfulness (and mindfulness interventions) generally [44–48], as well as underlying the relationship between mindfulness and prosocial behaviour more specifically [6,7,43,49]. That is, the present-moment attention and accepting orientation to experience constituting mindfulness [23] is thought to impact how we understand and interact with ourselves, and, through this, other outcomes of interest.

One candidate self-related process especially relevant to the relationship between mindfulness and compassion is self-focused attention. Self-focused attention is attention paid to self-referring, internally generated information, such as thoughts and emotions about the self [50], which can be either adaptive or maladaptive [51]. Maladaptive self-focused attention, or self-absorption [51], is self-focused attention that is sustained, excessive, and inflexible [50]. Mindfulness is conceptually related to lower levels of maladaptive self-focused attention because when one is more mindful, they also hold a more accepting attitude toward any self-related inner experiences that arise, which could allow for easier disengagement and less sustained and inflexible focus on the self (i.e., less self-absorption) [52], and this has been borne out empirically [24,53–56]. Further, lower maladaptive self-focused attention has also been found to mediate the relationship between mindfulness and positive intrapersonal psychological outcomes such as lower social anxiety [54] and greater self-efficacy [56]. While this has been tested as a mediator of intrapersonal outcomes, research is needed to understand if it acts as a mediator of interpersonal outcomes. Less self-absorption could free up attentional resources which, if deployed to present-moment experience, could allow for more attuned interaction with one's environment, including other people. Therefore, if higher mindfulness is associated with more compassionate behaviour, reduced self-absorption may mediate this relationship. Research is needed to understand if this is the case. Thus, our second hypothesis pertains to the role of self-absorption as a possible mediator underlying the relationship between mindfulness and compassion.

Whether or not mindfulness increases compassionate behaviour through lower self-absorption, however, may also depend on how motivated a person is to prioritize others' well-being. More mindfulness and lower self-absorption may increase the capacity for compassionate responding, but one must also value behaving compassionately in order to do so. One's self-construal, or the degree to which one identifies as being independent from or interdependent with others [57], may impact how important it is to act compassionately; for example, having higher interdependent self-construal has been related to having more compassionate goals [58]. Work on this so far is limited, and while some research suggests that mindfulness is positively related to interdependent self-construal [59–61], other work suggests that interdependent self-construal moderates the relationship between mindfulness and prosocial behaviour, such that mindfulness is positively associated with prosocial behaviour in those with higher interdependent self-construal [61–63]. These moderation effects suggest that greater mindfulness may enhance congruence between priorities and behaviour, and it is

plausible that mindfulness enhances compassionate behaviour especially in those for whom others' well-being is important. Therefore, mindfulness may be related to higher compassion through lower self-absorption more so in those with higher interdependent self-construal, and less so in those with lower interdependent self-construal; this is our third hypothesis.

Together, this study utilizes experience-sampling methods with romantic couples to examine the relationship between mindfulness and compassion in daily life. We hypothesized that mindfulness will positively relate to compassion towards one's partner in daily life, that this relationship will be mediated by lower self-absorption, and that this mediation will be moderated by interdependent self-construal.

## Methods
### Participants
We used the shiny app for power analysis for the Actor-Partner Interdependence Model (APIM) by Ackermann and Kenny[64]. Averaging estimates with effect sizes from our pilot study as well as previous literature ($\beta_{MeanActor} = 0.26$, $\beta_{MeanPartner} = 0.014$) as well as accounting for the increased power of a longitudinal study[65], we determined that a final sample size of $n = 100$ dyads would be sufficient, and aimed to oversample to $n = 135$ dyads to account for possible attrition.

We recruited a sample of participants from the United Kingdom via Prolific Academic. We invited participants who were over the age of 18, fluent in English, were willing to download an app, had at least an 80 percent approval rate, and were in a heterosexual, romantic relationship with a partner with whom they were cohabiting and who was also willing to participate in the study. No inclusion criteria were specified regarding relationship length.

Our total sample comprised 244 participants. Participants were between the ages of 20 and 79, with an average age of 40.33 years (SD = 11.93). They were 71% white, 9% Asian, 9% Black, 2% mixed, 1% other, and 8% not reported. Participants were 50% female, 49% male, and 1% not reported; sex was determined through self-reported demographic data provided by Prolific. Sixteen percent of participants reported currently practicing meditation. The dyads had a mean relationship length of 15.01 years (SD = 10.76) and a mean cohabitation length of 13.3 years (SD = 10.62). However, not all participants completed both the full-length trait measurements and the daily diary measurements. For our primary data of interest, the daily diary measurements, we collected 2705 daily diary observations from 230 participants belonging to 116 dyads (114 dyads with both members contributing). After preregistered exclusions due to values of the main variables being ±3 SDs from the mean (1 participant) and failure of attention check (7 surveys), we were left with 2698 daily diary surveys. Participants completed an average of 11.7 (SD = 3) daily diaries. We also collected full-length trait measurements, which were completed on the first day of the study. For these, we collected data from 206 participants belonging to 112 dyads (94 dyads with both members contributing).

### Procedure
A pilot study had been conducted before the general data collection with 9 dyads to assess the suitability of the study structure and measurements. For the main study, both trait questionnaires and daily diary measurements were collected using Expiwell (https://www.expiwell.com). After recruitment on Prolific, participants were provided with instructions regarding how to download the Expiwell app and begin the study. Participants were able to begin the study once both partners had successfully input the study code into the Expiwell app and additionally input a unique group code that both partners created together to identify them as a couple.

The study lasted for 15 days on the Expiwell app. On the first day, participants had until 8:59 p.m. to complete trait questionnaires. Following this, each evening, participants received a notification at 9:00 p.m. to complete a daily diary survey. They had 4 h and 59 min in which to do so. Participants were asked to complete 10 out of the 15 available days. This was to allow flexibility for participants to skip days on which they had not interacted with their romantic partner.

Participants were paid a base rate of £6/hour. Additionally, to encourage participation on the same days, participants were paid at a rate of £9/h for days on which both partners completed the daily survey. To encourage full study participation, participants were paid an additional 3.8£ if both partners completed 10 daily diary surveys on the same days as well as the trait questionnaires.

Participants were asked not to discuss their answers with their romantic partner until the entire study was completed. Data was collected between October 7th, 2024 and January 16th, 2025. All participants gave informed consent before participating in the study. The study was approved by the Ethics Committee of the University of Bamberg.

### Measurements
Each construct measured in this study (mindfulness, self-absorption, interdependent self-construal, and compassion both from one's partner and from one's self) was measured with both a retrospective trait questionnaire and with daily diary questions. We selected daily diary questions as our experience-sampling approach to facilitate partner reporting, because if participants were asked to report on, for example, the last hour rather than the last day, they may not have interacted with their partner within that time frame. As described below, daily diary measurements were decomposed into within-person and between-person measurements. The within-person component captures state-like fluctuations, whereas the between-person component captures trait-like individual differences in average daily levels[66]. For ease, we refer to within-person as state, and between-person as trait. It is important to note, however, that these "trait" measurements differ from the full-length, retrospective trait questionnaires that participants completed at the beginning of the study. Following the completion of these questionnaires, participants began the daily diary surveys. Each daily diary survey began by asking participants how much time they had spent with their partner that day and when they had interacted. If they had not spent any time with their partner, they were asked to exit the survey and complete it another day. Otherwise, they were instructed to complete the daily diary measurements.

The daily diary questions used were adapted from published scales to fit the daily diary as well as the dyadic structure of the study; the full-length, retrospective trait questionnaires were unchanged from the original published instruments. The response scales for the daily diary surveys were adapted so that they were consistent with each other and so that they were clearly visible with the Expiwell app display (each daily diary question was answered with a 1–6 Likert-type response scale). Adjustments to daily diary questions were also based on feedback received during the pilot study, as suggested by Shrout and Lane[67].

The number of questions asked was limited to reduce participant burden and increase likelihood of continued survey participation (notably, average completion was above the target of 10 daily diary surveys; psychometric properties of shortened scales have been found to perform adequately[68–70]. We planned to combine items that reached an interitem correlation of 0.3 (following recommendations by Hajjar[71] and Clark and Watson[72]). Combined items were summed and therefore had a response scale of 2–12, while single items maintained the response scale of 1–6.

To assess construct validity, we report the correlations between the retrospective trait and the daily diary measurements. We expect differences between these measures, since daily diary measurements are contextualized to the daily life scenarios we are interested in, and are less biased by recall errors when participants are asked to report aggregations of experiences over longer time periods[73]. The correlations between the trait and daily diary measures reported below are within the expected range for correlations between trait and sampling measurements (e.g., see Fleeson and Gallagher[74]).

**Mindfulness.** The mindfulness questions were taken from the Multidimensional State Mindfulness Questionnaire[75] and were selected based on the two component conceptualization from Bishop et al.[23], in which the self-regulation of attention was measured with the question "During

time spent with my partner today, I sometimes did not stay focused on what was happening in the present" (reverse-scored) and orientation to experience was measured with "During time spent with my partner today, I accepted my thoughts/feelings without dwelling on them," ($r_{within}$ = 0.19 and $r_{between}$ = 0.41). Since the within-person correlation did not reach 0.3, we treated mindful attention and acceptance as separate in the analyses. We additionally measured trait mindfulness using the Cognitive and Affective Mindfulness Scale-Revised[76], which showed good internal consistency ($\alpha$ = 0.83). The correlation between trait and daily diary mean mindfulness was within the expected range at $r$ = 0.45.

**Self-absorption**. Self-absorption was measured using the items: "During time spent with my partner today, I was not able to stop thinking about myself," and "During time spent with my partner today, I felt like my partner was constantly evaluating me," from the Self-Absorption Scale[77] ($r_{within}$ = 0.31 and $r_{between}$ = 0.41). The full-scale trait measurement of the Self-Absorption Scale showed excellent internal consistency ($\alpha$ = 0.92). The correlation between trait and daily diary mean self-absorption was within the expected range at $r$ = 0.45.

**Self-construal**. Interdependent self-construal was measured with the items: "During time spent with my partner today, it was important for me to maintain harmony with them," and "During time spent with my partner today, my happiness depended on the happiness of my partner," from the interdependent subscale of the Self-Construal Scale[78] ($r_{within}$ = 0.32 and $r_{between}$ = 0.61). Trait measurement of interdependent self-construal showed acceptable internal consistency ($\alpha$ = 0.79). The correlation between trait and daily diary mean interdependent self-construal was within the expected range at $r$ = 0.49.

**Compassion from one's partner**. Compassion from one's partner was measured with the items: "During time spent with my partner today, they noticed and were sensitive to any unpleasant feelings that I experienced," and "During time spent with my partner today, my partner treated me with feelings of support, helpfulness and encouragement," from the "compassion from others" subscale of the Compassionate Engagement and Action Scales[9] ($r_{within}$ = 0.46 and $r_{between}$ = 0.60). The full-scale trait measurement of compassion from others showed excellent internal consistency ($\alpha$ = 0.95). The correlation between trait and daily diary mean ratings of compassion from one's partner was within the expected range at $r$ = 0.64.

**Compassion towards one's partner**. Compassion towards one's partner was measured with the items "During time spent with my partner today, I noticed and was sensitive to any unpleasant feelings that they experienced," and "During time spent with my partner today, I expressed feelings of support, helpfulness, and encouragement to them," which were from the "compassion to others" subscale of the Compassionate Engagement and Action Scales[9] ($r_{within}$ = 0.37 and $r_{between}$ = 0.53). The full-scale trait measurement of compassion towards others showed excellent internal consistency ($\alpha$ = 0.92). The correlation between trait and daily diary mean ratings of compassion towards one's partner was within the expected range at $r$ = 0.58.

For each of the compassion questions referring to unpleasant feelings, we included an N/A response option: "I did not experience any unpleasant feelings today/My partner did not show that they experienced any unpleasant feelings today."

We additionally measured variables such as relationship satisfaction (using the Relationship Assessment Scale[79]) and meditation practice for the purpose of future exploratory research, but did not include these variables in the current study.

## Statistical analyses
All analyses were conducted using R[80] and RStudio (version 2025.9.0.387)[81], using packages rio[82], dplyr[83], psych[84], and nlme[85].

We conducted analyses with both the daily diary as well as the full-length trait questionnaire data. The daily diary data was our primary interest, and we report the secondary full-length trait questionnaire analytic approach and results in the supplementary information. Within the daily diary data, to understand the relationships between the constructs of interest both within and between person, we decomposed our predictor variables into between-person means (which we also refer to as trait) and within-person deviations (which we also refer to as state) via person-mean centering. As noted above, our primary outcome of interest was other-rated compassion. However, since much previous work utilizes self-ratings of compassion, we also conducted self-ratings of compassion as a secondary outcome.

Due to the dyadic structure of the data, we planned to employ an Actor-Partner Interdependence Model (APIM), contingent on model comparison testing. The APIM statistically accounts for mutual influence within dyads by including predictors from both members (i.e., actor and partner effects)[86]. For each model, we first tested if the APIM structure was necessary, and then if the dyad members were distinguishable by sex, as suggested by Kenny and Ledermann[87]. Results of the model testing are available in Table S1 in the supplementary information. Based on these model comparisons, we selected indistinguishable APIM models (i.e., models that do not include sex as a distinguishing variable) to test our hypotheses.

Our daily diary data had a three-level structure, with daily observations nested within participants, and participants nested within dyads. To account for the nested structure of the data, we used a multilevel model with random intercepts for dyads, and random intercepts and slopes for participants. The random effects model for participants includes a random intercept and random slopes, along with covariances between random intercepts and slopes, but sets covariances between slopes to zero. This model incorporates between-participant variability in random effects while maintaining a parsimonious structure, with a parametrization that is consistent with the number of observations. Since our analysis was longitudinal, we included both between-person means (trait level) and within-person deviations (state level) in all models, in order to mitigate possible biasing of the estimates of state-level coefficients with the between-person associations among variables[66]. All tests were two-sided. Distributional assumptions were checked through examinations of residuals using, among other methods, the DHARMa package[88] in R.

To test self-absorption as a mediator underlying the relationship between mindfulness and compassion, we constructed a stacked multilevel model. In this approach, the mediator and outcome variables are stacked into a single response structure, allowing the mediator and outcome equations to be estimated simultaneously within one longitudinal mixed-effects model[89]. This allows bootstrapping the estimates of indirect effects in a manner that accounts for covariances between the 'a' and the 'b' components of the indirect effect. The indirect effects of the actor variables of interest were computed while controlling for the partner effects. The partner effects are marginal to actor effects, that is, the estimates of partner effects include their indirect effects through their relationships with actor effects. Bootstrap confidence intervals were obtained using bias-corrected and accelerated (BCa) bootstrapping using the method described in DiCiccio and Efron[90]. BCa intervals, in contrast with bias-corrected (BC) intervals, are particularly appropriate where the distribution of bootstrap replicates is likely to be far from normal and skewed[91], as expected in the bootstrap sampling of indirect effects. To test self-construal as a moderator of the relationship between self-absorption and compassion, we included a self-absorption by self-construal interaction term.

## Preregistration
This study (including research question, hypotheses, analysis plan, etc.) was preregistered on October 5th, 2024 under: https://osf.io/ejmkw/overview?view_only=14829c5a4add44d8856d4c3ea8c6af6e (project: osf.io/c6nxk).

# Results

Weighted means, weighted between-person standard deviations, within-person standard deviations and correlations of relevant variables are summarized in Table 1.

Consistent with the definition of compassion as a response to suffering, we analyzed compassion only on days when an opportunity for such a response occurred. Therefore, any N/A responses to the compassion questions ("I did not experience any unpleasant feelings today/My partner did not show that they experienced any unpleasant feelings today") were dropped from the analysis. Further, we only had partner-rated compassion outcome values for days on which both partners completed the daily diary survey. Therefore, the number of observations available for analyses with partner-rated compassion was $n_{obs} = 1333$, and the number of observations available for analyses with self-rated compassion as the outcome was $n_{obs} = 1332$.

## Hypothesis 1: Mindfulness is positively related to compassion

We found support for this hypothesis both within- and between-person, and with both partner- and self-ratings of compassion. However, the associations of the acceptance and the attention facets of mindfulness differed.

**Within person.** We found that within-person fluctuations of the acceptance facet of mindfulness positively predicted both partner- ($t(1048) = 3.09$, $B = 0.19$, SE = 0.06, $p = 0.002$, 95% CI[0.07, 0.31]) and self- ($t(1043) = 4.66$, $B = 0.27$, SE = 0.06, $p < 0.001$, 95% CI[0.16, 0.39]) ratings of compassion. However, we did not find evidence that within-person fluctuations of the attention facet of mindfulness were related to either partner- ($t(1048) = 0.80$, $B = 0.05$, SE = 0.06, $p = 0.425$, 95% CI[−0.08, 0.18]) or self- (($t(1043) = 1.31$, $B = 0.07$, SE = 0.06, $p = 0.190$, 95% CI[−0.04, 0.19]) ratings of compassion.

**Between person.** A different pattern was found at the between-person level. We did not find evidence that the acceptance facet of mindfulness was related to partner-ratings of compassion (t(88) = −0.40, $B = -0.06$, SE = 0.15, $p = 0.687$, 95% CI[−0.36, 0.24]). We found some evidence that it was related to self-ratings of compassion ($t(91) = 2.06$, $B = 0.26$, SE = 0.13, $p = 0.04$, 95% CI[0.01, 0.52]). However, we found stronger evidence that the attention facet of mindfulness was related to both partner- ($t(88) = 3.15$, $B = 0.52$, SE = 0.17, $p = 0.002$, 95% CI[0.19, 0.85]) and self- ($t(91) = 3.80$, $B = 0.52$, SE = 0.14, $p < 0.001$, 95% CI[0.25, 0.79]) ratings of compassion.

All results (including partner effects) can be found in Table S2 in the supplementary information.

## Hypothesis 2: The relationship between mindfulness and compassion is mediated by self-absorption

We found evidence supporting our mediation hypothesis at both the within- and between- person level, for both partner- and self-ratings of compassion,

and for both facets of mindfulness, though with some variation. We report the direct and indirect effects in text, paths in Figs. 1 and 2, and the full models (Table S3) and coefficient plots (Figures S1 and S2) in the supplementary information.

### Within person

Acceptance. The direct effect of acceptance on partner-rated compassion was non-significant at the within-person level ($c' = 0.07$, 95% BCa CI[−0.05, 0.20]). We found evidence for an indirect effect of acceptance on partner-rated compassion via self-absorption at the within-person level (ab = 0.06, 95% BCa CI[0.03, 0.08]). See Fig. 1a. The direct effect of acceptance on self-rated compassion was significant at the within-person level ($c' = 0.21$, 95% BCa CI[0.08, 0.32]). We found evidence for an indirect effect of acceptance on self-rated compassion via self-absorption at the within-person level (ab = 0.05, 95% BCa CI[0.03, 0.08]). See Fig. 1b.

Attention. Although the total effect of attention on compassion was non-significant at the within-person level (see Hypothesis 1), we found evidence for indirect effects via self-absorption. Specifically, the direct effect of attention on partner-rated compassion was non-significant at the within-person level ($c' = -0.003$, 95% BCa CI[−0.13, 0.14]), but we found evidence for an indirect effect of attention on partner-rated compassion via self-absorption at the within-person level (ab = 0.07, 95% BCa CI[0.04, 0.11]). See Fig. 1c. Similarly, the direct effect of attention on self-rated compassion was non-significant at the within-person level ($c' = -0.01$, 95% BCa CI[−0.12, 0.10]), but we found evidence for an indirect effect of attention on self-rated compassion via self-absorption at the within-person level (ab=0.07, 95% BCa CI[0.04, 0.10]). See Fig. 1d.

### Between person

Acceptance. The direct effect of acceptance on partner-rated compassion was non-significant at the between-person level ($c' = -0.11$, 95% BCa CI[−0.45, 0.20]). We found weak evidence for an indirect effect of acceptance on partner-rated compassion via self-absorption at the between-person level (ab=0.03, 95% BCa CI[0.002, 0.06]). See Fig. 2a. The direct effect of acceptance on self-rated compassion was non-significant at the between-person level ($c' = 0.22$, 95% BCa CI[−0.06, 0.52]). We did not find evidence for an indirect effect of acceptance on self-rated compassion via self-absorption at the between-person level (ab = 0.03, 95% BCa CI[0.00, 0.06]). See Fig. 2b.

Attention. The direct effect of attention on partner-rated compassion was non-significant at the between-person level ($c' = 0.24$, 95% BCa CI[-0.07, 0.63]). We found evidence for an indirect effect of attention on partner-rated compassion via self-absorption at the between-person level (ab = 0.21, 95% BCa CI[0.11, 0.30]). See Fig. 2c. The direct effect of attention on self-rated compassion was significant at the between-person level ($c' = 0.35$, 95% BCa

## Table 1 | Descriptive Statistics and Correlations

| # | Variable | M | BP SD | WP SD | 1 | 2 | 3 | 4 | 5 | 6 |
|---|----------|---|-------|-------|---|---|---|---|---|---|
| 1 | Mindfulness (acceptance) | 8.76 | 1.98 | 2.37 | — | 0.19*** | −0.28*** | 0.22*** | 0.16*** | 0.17*** |
| 2 | Mindfulness (present-moment attention) | 9.23 | 1.81 | 2.22 | 0.41*** | — | −0.28*** | 0.08*** | 0.03 | 0.04 |
| 3 | Self-absorption | 3.81 | 1.30 | 1.44 | −0.40*** | −0.70*** | — | −0.18*** | −0.21*** | −0.23*** |
| 4 | Self-construal | 8.60 | 2.05 | 1.53 | 0.18** | 0.25*** | −0.22*** | — | 0.20*** | 0.35*** |
| 5 | Compassion (other-rated) | 8.01 | 1.99 | 1.97 | 0.08 | 0.24*** | −0.30*** | 0.26*** | — | 0.36*** |
| 6 | Compassion (self-rated) | 8.41 | 1.78 | 1.77 | 0.19** | 0.33*** | −0.29*** | 0.49*** | 0.52*** | — |

** $p < 0.01$, *** $p < 0.001$. "BP SD" indicates between-person standard deviation; "WP SD" indicates within-person standard deviation. Within-person correlations above the diagonal; between-person correlations below the diagonal (between-person variance includes both individual and dyad-level components). Weighted means and between-person standard deviations. Mindfulness (attention) and (acceptance) rescaled from 1–6 for this table by doubling values to match the scales of the other variables (2–12).

a)

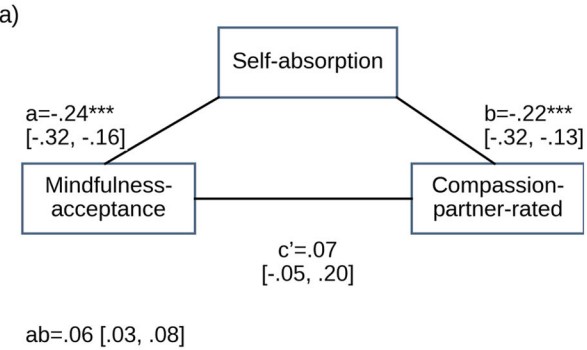

ab=.06 [.03, .08]

b)

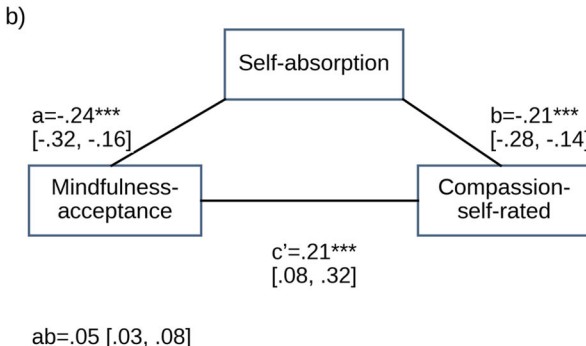

ab=.05 [.03, .08]

c)

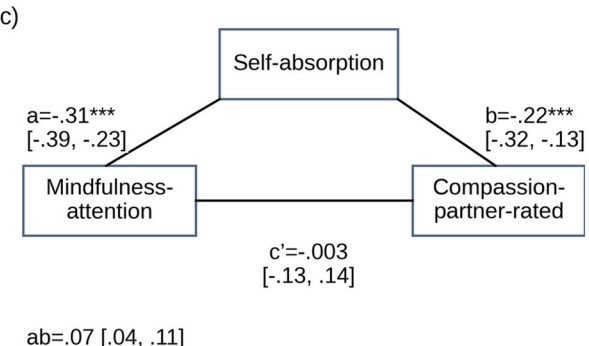

ab=.07 [.04, .11]

d)

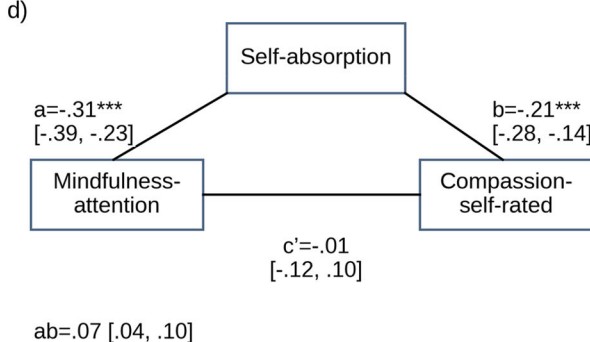

ab=.07 [.04, .10]

**Fig. 1 | Within-Person Mediations of Mindfulness–Compassion Associations by Self-Absorption.** Note. For mediation figure, * $p < 0.05$, ** $p < 0.01$, *** $p < 0.001$. 95% BCa confidence intervals reported. This figure shows the mediation of the relationship between mindfulness and compassion by self-absorption at the within-person level. The predictors are either mindful acceptance (**a, b**) or mindful attention (**c, d**). The outcomes are either partner-rated compassion (**a, c**) or self-rated compassion (**b, d**). The number of daily diary observations was $n_{obs} = 1333$ for partner-rated compassion and $n_{obs} = 1332$ for self-rated compassion.

CI[0.05, 0.65]). We found evidence for an indirect effect of attention on self-rated compassion via self-absorption at the between-person level (ab = 0.20, 95% BCa CI[0.12, 0.27]). See Fig. 2d.

### Hypothesis 3: The mediation of the relationship between mindfulness and compassion by self-absorption is moderated by interdependent self-construal

We did not find evidence to support the hypothesis that interdependent self-construal moderated the mediation of the relationship between mindfulness and compassion by self-absorption. We began by testing whether self-construal moderated the relationship between self-absorption and compassion. Because we did not find evidence for this moderation for our primary outcome of partner-rated compassion, we did not proceed with the moderated mediation analysis.

**Within person.** We did not find evidence at the within-person level for an interaction between self-absorption and self-construal when predicting partner-rated compassion ($t(1042) = -1.16$, $B = -0.02$, SE = 0.02, $p = 0.246$, 95% CI[−0.06, 0.02]). We found weak evidence for an interaction between self-absorption and self-construal when predicting self-rated compassion ($t(1037) = 2.01$, $B = 0.03$, SE = 0.02, $p = 0.044$, 95% CI[0.001, 0.07]).

**Between person.** We did not find evidence at the between-person level for an interaction between self-absorption and self-construal when predicting either partner- ($t(82) = -0.41$, $B = -0.02$, SE = 0.05, $p = 0.680$, 95% CI[−0.11, 0.07]) or self- ($t(85) = 1.51$, $B = 0.05$, SE = 0.04, $p = 0.136$, 95% CI[−0.02, 0.13]) ratings of compassion.

All results (including partner effects) can be found in Table S4 in the supplementary information.

### Exploratory analyses
We conducted additional exploratory analyses that are reported in the supplementary information.

### Discussion
As mindfulness grows in popularity, it is crucial to use methods with greater ecological validity to examine whether findings based on retrospective trait measures or experimental designs generalize to people's daily lives. To address this gap, we employed a daily diary design, which allowed us to capture nuanced differences in how mindfulness and compassion relate at the state level (within persons) and at the trait level (between persons). At the within-person, or daily state level, we were able to investigate how changes in mindfulness day to day varied with compassion towards one's partner day to day. We found support for an association between mindful acceptance and compassion at this level. At the between-person, or trait level, we were able to investigate how tendencies to experience mindful states related to tendencies to be compassionate to one's partner. At this level, we found support for an association between mindful attention and compassion. Further, we used partner ratings of compassion to enhance both the validity and ecological applicability of these findings by reducing reliance on single source self-report and integrating an interpersonal perspective. Because compassion serves an important social function, partner ratings also help clarify how mindfulness relates to compassion in everyday interpersonal contexts. Results from this study support the existence of a relationship between mindfulness and compassion in real-life contexts, and reveal variation in terms of how facets of mindfulness relate to compassion both within and between persons. As mindfulness research expands into domains such as social behaviour, it becomes increasingly important to identify the mechanisms underlying these relationships. We found support for self-absorption as a possible underlying mechanism, but did not find evidence

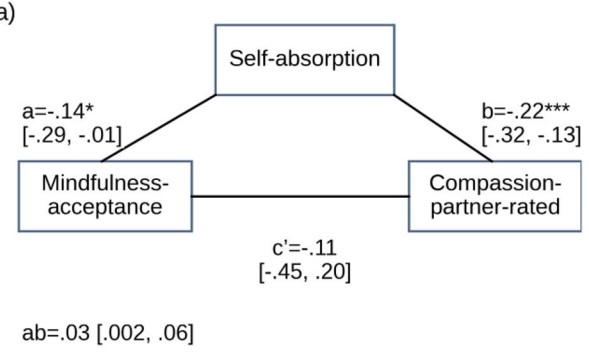

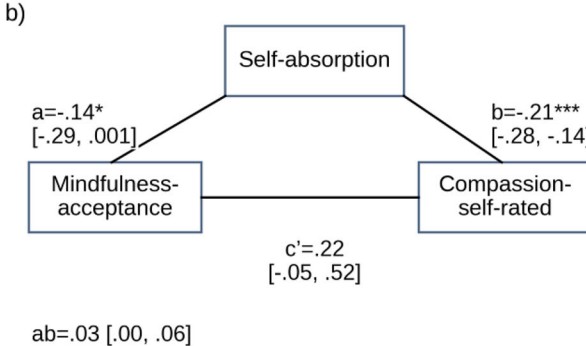

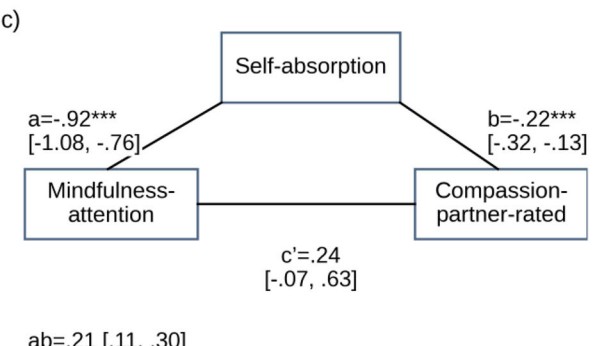

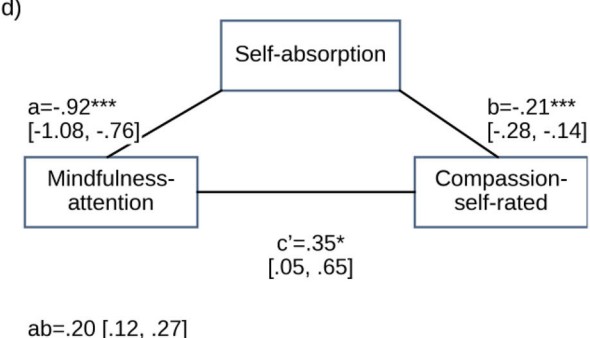

**Fig. 2 | Between-Person Mediations of Mindfulness–Compassion Associations by Self-Absorption.** Note. For mediation figure, * $p < 0.05$, ** $p < 0.01$, *** $p < 0.001$. 95% BCa confidence intervals reported. This figure shows the mediation of the relationship between mindfulness and compassion by self-absorption at the between-person level. The predictors are either mindful acceptance (**a**, **b**) or mindful attention (**c**, **d**). The outcomes are either partner-rated compassion (**a**, **c**) or self-rated compassion (**b**, **d**). The number of daily diary observations was $n_{obs} = 1333$ for partner-rated compassion and $n_{obs} = 1332$ for self-rated compassion.

that this relationship was moderated by self-construal. Each of these findings is discussed further below.

### Mindfulness and compassion in daily life

This study supports the generalizability of previous research findings on the relationship between mindfulness and compassion into people's daily lives, as we find support for a relationship between mindfulness and compassion both within- and between-person. We expected to be able to combine the mindfulness facets of acceptance and attention. However, these facets of mindfulness did not reach the preregistered interitem correlation threshold at the within-person level. Therefore, we treated these facets as separate.

We found differing patterns at the within- versus between-person levels. At the within-person level, we found an association between the acceptance facet of mindfulness and compassion, but not between the attention facet of mindfulness and compassion. This means that, as one's mindful acceptance fluctuated from day to day, there were corresponding changes in their compassion, both as perceived by oneself and by one's partner. However, we did not find evidence for an association between daily changes in one's mindful attention and compassion. We also found greater within-person variability in acceptance than in attention. At the between-person level, an almost inverse pattern was found, in which there was an association between mindful attention and compassion, but weaker evidence for an association between mindful acceptance and compassion. In other words, if one person is more mindfully attentive on average, then they are also more compassionate on average, both as perceived by oneself and by one's partner. However, we found relatively weak evidence that being more mindfully accepting on average was related to being more compassionate on average as perceived by oneself, and did not find evidence that this held for how compassionate one was on average as perceived by their partner. Overall, then, one's own fluctuations in daily state acceptance (but not

attention) are associated with fluctuations in one's compassion, and one's general, or trait, levels of attention (but less so acceptance) are associated with one's general levels of compassion.

As noted, day-to-day variability in mindful acceptance was associated with day-to-day variability in compassion. This suggests that an individual's acceptance fluctuates at a daily level in ways that meaningfully relate to outcomes that are important to people's lives and relationships. However, mindful acceptance had a weaker relationship at the between-person level. This may signify that it is not acceptance in general that is important, but acceptance at appropriate times (i.e., those in which compassion is relevant).

Conversely, the evidence supporting an association between between-person attention and compassion suggests that a general tendency to attend to the present moment may be associated with the capacity to regularly notice environmental cues that elicit compassion, resulting in stronger perceived compassion overall. However, present-moment attention may not vary enough within-person from one day to another to noticeably relate to compassion at the daily level, or in other words, mindful attention may reflect a more stable orientation. We investigated the possibility of a ceiling effect but did not find evidence for this (see supplementary information). Another consideration is that, when one is not mindfully attentive, what they are otherwise attending to may be relevant. Previous research on mind wandering and caring for others found within-person associations such that mind wandering to something negative or neutral predicted less caring for others behaviour, but not when the mind wandered to something positive[92]. In line with this result, we did find an indirect relationship between mindful attention and compassion through self-absorption at the within-person level. This evidence may suggest that daily fluctuations in attention are related to compassion when associated with fluctuations in specific other kinds of cognition, such as self-absorption. This issue is discussed further below. A further important note is that it is possible that fluctuations in

attention throughout the day (rather than day to day) may still directly relate to compassion, and more frequent assessment throughout the day may uncover such associations. Further research is needed to test this.

Monitor and Acceptance Theory[93] posits that acceptance serves an important emotion regulation mechanism modulating the effects of present-moment attention. Thus, whether mindful attention is associated with compassion may be more likely if one can also emotionally regulate their response to the present moment through acceptance. Following this line of thought, we exploratorily tested interactions between attention and acceptance, to see if, for example, state, or daily fluctuations in acceptance were more strongly related to compassion in those with higher trait, or average levels of present-moment attention. We did not find support for this idea (see supplementary information), but the dataset may have been too small and too homogeneous to show such effects. We thus recommend such an investigation as an avenue for future research.

Together, these findings highlight the importance of investigating mindfulness as both a state that fluctuates through time and as a trait. For certain outcomes such as compassion, specific facets of mindfulness may meaningfully vary in tandem with these outcomes from day to day—patterns that may be obscured when considering only trait-level associations. Conversely, other facets may show stronger associations with outcomes at the trait level. Thus, longitudinal designs are crucial for uncovering the nuanced ways in which different facets of mindfulness relate to important outcomes such as compassion.

### Mindfulness and compassion as mediated by self-absorption
We found support for the hypothesis that self-absorption mediates the relationship between mindfulness and compassion. This finding is in line with more general theories of the role of self-related processes, which posit that mindfulness is associated with prosocial outcomes via differences in self-related processing[6,7,43,49]. We found this for both the acceptance and attention facets of mindfulness, and for both self- and partner-ratings of compassion. We also found this at both within- and between-person levels for attention, but only at the within-person level for acceptance. Therefore, when one has higher mindful acceptance or attention than one usually does, they also have lower self-absorption than they usually do, and this mediates the relationship with higher than usual compassion. Further, those who have higher mindful attention generally also have lower self-absorption generally, and this mediates the relationship with higher compassion generally; however, this was not found for acceptance. In terms of the relationship between mindful attention and self-absorption at the between-person level, we found a strong correlation, which may raise concerns about construct distinctness. However, mindful attention and self-absorption represent dissociable constructs theoretically. For instance, it is possible to be low on mindful attention but to not be focused on the self (for example, one could be distracted by thinking about an interesting topic). The dissociability of these constructs is further supported by the smaller correlation at the within-person level. If mindful attention and self-absorption reflected different ends of one dimension, we would expect to find a similarly large correlation at both levels, but this was not supported by the evidence.

Interestingly, and as noted above, although we did not find evidence for a direct relationship between within-person mindful attention and compassion with our previous hypothesis, we did find evidence for an indirect effect through self-absorption, highlighting the importance of investigating underlying mechanisms for understanding these relationships. As discussed above, such findings can provide a more nuanced understanding of the relationship between mindfulness and compassion. Overall, these results support the idea that mindful attention and acceptance may make it easier to disengage from rigid, self-referential processing, possibly freeing up attentional resources to notice and respond to interpersonal cues eliciting compassion.

### Interdependent self-construal as a moderator
We did not find evidence to support the idea that interdependent self-construal moderates the mediation by self-absorption as described above.

This may be due to a few reasons. An important difference between the present research and previous research is that previous studies did not investigate a moderation of the mediation of self-absorption, but rather a moderation of mindfulness itself. To address this issue in an exploratory fashion, we tested self-construal as a moderator of the relationship between mindfulness and compassion (without self-absorption). We did not find evidence to support this (results reported in the supplementary information). Another difference with previous research is that the moderating effect of self-construal has been found either in an experimental setting, following a mindfulness[94] or self-construal[63] induction, or with trait measurements[61,62]. Therefore, research was needed to study this moderation with experience-sampling methods. It is possible that, if a moderation effect exists in daily life, it is smaller in this context than with other methods, and we may have been underpowered to detect such an effect. Furthermore, such an effect may be stronger and thus more detectable if participants were to be sampled from populations that vary more in terms of interdependent self-construal, for example, if participants are sampled from both individualistic and collectivistic cultures.

In addition to past research showing a moderation effect, there is other research suggesting that mindfulness is positively related to interdependent self-construal[59–61]. Since interdependent self-construal is itself related to having compassionate goals[58], we also conducted exploratory analyses of self-construal as a mediator between mindfulness and compassion. We found some support for this in terms of self-ratings of compassion (results reported in the supplementary information). However, as this was exploratory, these results should be interpreted primarily as motivating potential future confirmatory research into the possible mediating role of interdependent self-construal on the relationship between mindfulness and compassion. Overall, then, as discussed in the introduction, investigations into mindfulness, self-construal, and prosocial behaviour are somewhat limited, and future research should continue to investigate this question to provide greater clarity.

### Limitations and future research directions
A limitation of this study is that single item measures were used for mindful acceptance and mindful attention. We had intended to combine these items into one mindfulness measurement, but they did not reach the preregistered interitem correlation threshold at the within-person level. However, the associations between these measures and the full-length trait measurement were found to be within the expected range. Further, single-item experience-sampling measurements have been shown to have good concurrent and predictive validity[95]. Still, to be able to assess internal reliability, it would be advisable for future research examining within-person effects to include more items per mindfulness facet measured, especially if the facets are dissociable when considering within- versus between-person effects. Another limitation is that we cannot infer causation between mindfulness, self-absorption, and compassion. Future research can add to the robustness of these findings by investigating whether or not self-absorption mediates a link between experimentally induced increases in mindfulness and compassion, through, for example, the use of a mindfulness-based intervention. An additional limitation is that our sample was limited to the United Kingdom. While our sample does allow for broader generalizability than most research in this area, since it sampled from the general adult population rather than from a student population, as has been the case in the majority of research investigating mindfulness and social behaviour[7], these results still may not generalize to other countries or cultures. Further, the dyads in our sample were heterosexual romantic couples. This limitation could be addressed by future work investigating if these findings generalize to couples within the LGBTQIA+ community. Beyond romantic couples, future work is also needed to understand the generalizability of these findings to how mindfulness and compassion relate in other interpersonal contexts, for example between strangers, colleagues, or friends.

Finally, future research should dive deeper into the relational dynamics of mindfulness, self-absorption, and compassion. In our statistical analyses, we used the APIM to properly account for the dyadic structure of the data.

However, we did not intend to investigate partner effects as part of our research question. Future research can utilize the partner effects found in this study (reported in the supplementary information) to aid in formulating novel hypotheses concerning these dynamics. For example, when partner-rated compassion was the outcome, partner mindfulness was also controlled for due to the structure of the APIM. This relationship between partner mindfulness and partner-compassion ratings, however, could also be used in formulating future hypotheses about how mindfulness relates to perceptions of compassion. That is, there may be associations between one's own mindfulness and one's perceptions of their partner's compassion. Exploratory analyses (reported in the supplementary information) also revealed further possible research directions and important considerations. For example, mindfulness predicted being more likely to report that no unpleasant feelings were experienced on a specific day, both for oneself as well as for one's partner. Mindfulness therefore may be associated with the dynamics of how these interactions develop and how often compassion is needed or considered an appropriate response in the first place. Additionally, future research may choose to differentiate between compassionate noticing and compassionate acting, as exploratory analyses revealed that mindfulness may be more relevant to others' perceptions of compassionate acting as compared with compassionate noticing. Lastly, future research should consider including time as a covariate. Though the focus of this study was not on the dynamics between these variables over time, and including the day of the survey as an exploratory covariate did not change the reported pattern of results, it did reveal that compassion ratings for both self and partner decreased as participants progressed through the study. It is possible that regularly rating mindfulness and compassion may have affected personal experiences and interactions.

## Conclusions

This study used daily diary methods and partner-ratings of compassion to investigate if previous work on the relationship between mindfulness and compassion generalizes to daily life. We found support for this hypothesis. The use of partner-ratings of compassion allowed us to better understand how these relationships play out in an important, real-world context, enhancing the relevancy of our findings. Further, this longitudinal analysis allowed us to disentangle within- and between-person associations, revealing nuance in how mindfulness relates to compassion. Daily fluctuations in acceptance were linked to day-to-day fluctuations in compassion, whereas average levels of attention were associated with compassion at the between-person level. This may signify that mindful acceptance may be most relevant when it occurs at times in which compassion is appropriate, and mindful attention may reflect a more stable orientation. We also investigated possible mechanisms and moderators related to the self. We found support for the idea that self-absorption mediates the relationship between mindfulness and compassion, suggesting that mindfulness may be associated with compassion partly through lower self-absorption. However, we did not find support for the idea that this relationship is moderated by interdependent self-construal. Together, these findings highlight conditions under which individuals may have the psychological resources to respond compassionately to their partners and thus offer potential avenues for interventions and couple counselling.

## Data availability

Data can be found under the associated OSF project: https://doi.org/10.17605/OSF.IO/C6NXK.

## Code availability

Analytic code can be found under the associated OSF project: https://doi.org/10.17605/OSF.IO/C6NXK.

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

## Acknowledgements

The authors would like to thank Barnaby Crook for his helpful comments on a draft of this manuscript, as well as the team at the Chair of Personality Psychology and Psychological Assessment at the University of Bamberg for feedback and discussion at various points throughout this project. We thank the pilot participants for their participation. We would also like to thank the statistical consulting service at York University for their assistance.

## Author contributions

T.W.: conceptualization, funding acquisition, study design, data collection, statistical analyses, writing – original draft preparation, and writing – review and editing; G.M.: statistical analyses, writing – original draft preparation for describing the mediation in the Methods section, and writing – review and editing; A.S.: project supervision, funding acquisition, and writing – review and editing.

## Funding

We thank the Standing Commission for Research and Young Scientists at the University of Bamberg for funding this project. The funders had no role in study design, data collection and analysis, decision to publish or preparation of the manuscript. Open Access funding enabled and organized by Projekt DEAL.

## Competing interests

The authors declare no conflict of interest.
