## [Transparent Peer Review file · Communications Psychology]

Higher Mindfulness and Lower Self-Absorption Predict Greater Compassion within Romantic Relationships

Corresponding Author: Ms Theresa Waclawek

Version 0:

Decision Letter:

Dear Ms Waclawek,

Thank you for your patience during the peer-review process. Your manuscript titled "Mindfulness, the Self, and Compassion: A Daily Diary Study" has now been seen by 3 reviewers, and I include their comments at the end of this message. They find your work of interest but raised some important points. We are interested in the possibility of publishing your study in Communications Psychology, but would like to consider your responses to these concerns and assess a revised manuscript before we make a final decision on publication.

We therefore invite you to revise and resubmit your manuscript, along with a point-by-point response to the reviewers. Please highlight all changes in the manuscript text file.

Editorially, we consider it critical that you address the conceptual and methodological concerns raised by the reviewers, especially regarding the conceptualization of mindfulness and methodological concerns regarding measurement and mediation analyses.

As you revise the manuscript in response to these issues, please also implement all requests in the attached Mandatory Revision Requests document. All requirements listed in this document need to be fully met, or the work will be returned to you for further revisions without peer review. This workflow is in place to increase the likelihood that the paper will be accepted for publication. It reduces the number of rounds of revision (and review) and ensures that the reviewers vet a version of the article that is compliant with journal policies. If you have any questions regarding the required revisions, please contact the journal prior to resubmission to avoid a negative outcome.

Please submit the following items:

- Revised manuscript
- Point-by-point response to the referees' comments
- Mandatory Revision Requests Table (attached).
- Cover letter (as a separate document)

via this link: Link Redacted .

** This url links to your confidential home page and associated information about manuscripts you may have submitted or are reviewing for us. If you wish to forward this email to co-authors, please delete the link to your homepage first **

Best regards,

Jennifer Bellingtier

Jennifer Bellingtier, PhD
Senior Editor
Communications Psychology

on behalf of
Hannah Hao, PhD
Editorial Board Member
Communications Psychology
orcid.org/0000-0002-3342-9132

REVIEWER EXPERTISE:
mindfulness, relationships, dyads

REVIEWER REPORTS:

Reviewer #1 (Remarks to the Author):

This is a well-conducted research study employing the daily diary method, which will inform thinking in the field. While the paper includes thorough research findings, the below may be helpful to consider in revisions to strengthen the paper:

Abstract: It would be important to add further details on the participant demographics in the abstract, such as age, gender, length of relationship, sexual orientation, etc. It would also be helpful to include the statistics of the main findings in the abstract. When describing "greater mindfulness", be more specific on whether you are referring to trait or state mindfulness here.

Introduction: Good to have introduced compassion and mindfulness, but you need to be much more specific and explain the difference between state and trait mindfulness, the theoretical background of this, and include research supporting their respective effectiveness. A reference is missing on recall bias for mindfulness when recorded retrospectively. Is there any empirical evidence as to why the daily diary method is an appropriate approach to use here, i.e. has this research been completed with other/similar constructs in the past? If so, it would be important to add this here to hypothesize why this may also be the case for mindfulness and compassion constructs. Do not include analysis plans in the introduction section, this needs to be in the methods section.

Methods: Participants and procedures are generally well outlined in the methods including relevant detail. When adjusting the response scales, was any psychometric analysis completed to account for difference in scoring of these scales? Was factor or Rasch analysis completed to ensure scale validity and/or differential item functioning with the new Likert scoring? If not, it would be important to complete these psychometric analyses to ensure internal consistency of the different scales. This is reported very briefly but needs to be made much more explicit in the manuscript, tables can be added in supplementary materials.

Results: Generally, well reported, and interesting results! Consider including leading zeros throughout. It would have been interesting to consider whether relationship length or age influenced these findings, consider repeating analyses where age and relationship length is controlled for.

Discussion: Similar to the introduction, more nuance is needed with regards to trait vs. state mindfulness and consolidate your findings with theory and previous research. It would be important to include further limitations of your methods in the limitations section of the discussion, such as limits of your findings with regards to generalisability, i.e. for couples who are part of the LGBTQI+ community, and what else future research needs to focus on in this respect.

Writing in general: there are some typographical errors and incorrect references (e.g. at times, page numbers are missing for direct quotes), and some references that are very dated. Consider updating this throughout.

Reviewer #2 (Remarks to the Author):

This preregistered daily diary study examines within- and between-person associations among mindfulness, self-absorption, and compassion in romantic couples. Using a dyadic design and partner-reported compassion, the authors provide an important contribution to the literature on mindfulness and social functioning by moving beyond global self-reports and laboratory paradigms to everyday relational contexts. The findings offer a nuanced account of how distinct facets of

mindfulness operate differently across levels of analysis and suggest self-absorption as a potential mechanism linking mindfulness to compassion. Below I outline several major and minor points that I believe would strengthen the conceptual clarity, methodological transparency, and interpretability of the findings.

Major Comments (Conceptual and Analytic)

1. Social desirability and partner reports

The manuscript suggests that partner-reported compassion may mitigate social desirability bias inherent in self-report. While partner reports offer a valuable complementary perspective, I encourage more caution in this claim. Partner reports are also susceptible to social desirability, shared method variance, and relationship-specific biases, particularly in close dyads. It may be more appropriate to frame partner reports as capturing perceived compassion in close relationships, rather than as a cleaner or less biased alternative to self-report.

2. High overlap between mindfulness attention and self-absorption at the between-person level

The between-person correlation between mindful attention and self-absorption ($r \approx -.70$) is strikingly large. This raises concerns about construct overlap and the interpretability of the between-person mediation model. It would be helpful for the authors to discuss whether these constructs are conceptually and empirically distinct at the trait level, or whether they may reflect opposite poles of a shared underlying dimension. Additional justification (or sensitivity analyses) would strengthen confidence in the mediation findings at the between-person level.

3. Mindfulness and perception of partner compassion

An interesting alternative pathway not addressed in the manuscript is whether individuals higher in mindfulness perceive their partners as more compassionate, independent of their own compassionate behavior. Mindfulness may enhance sensitivity to social cues or positive reinterpretation of partner behavior. Examining associations between one's own mindfulness and one's ratings of partner compassion could provide useful context and help disentangle behavioral effects from perceptual ones.

4. Divergent within- and between-person effects

The reversal of effects across levels of analysis—where mindful acceptance predicts compassion within persons, but mindful attention predicts compassion between persons—is one of the most intriguing findings of the paper. However, this pattern is not discussed in depth. Greater theoretical elaboration is needed to explain why these facets of mindfulness operate differently across levels. For example, acceptance may be more relevant for momentary interpersonal responsiveness, whereas attention may reflect a stable interpersonal orientation.

Relatedly, the compassion measure includes multiple items that may map differentially onto attention versus acceptance. Exploratory analyses examining item-level or subscale-level associations could help clarify whether specific components of compassion are driving the observed effects.

Minor Comments

1. Relationship demographics

Please clarify the relationship-related inclusion criteria (e.g., relationship length, cohabitation status) and consider reporting additional descriptive information about the couples.

2. Mindfulness practice

If available, it would be useful to report whether participants engaged in regular mindfulness-related practices (e.g., meditation, prayer), or to note explicitly if this information was not collected.

3. Response scale discrepancy

The daily diary response scale is described as ranging from 1–6, yet Table 1 reports means between 8–9. Please clarify this discrepancy.

4. Decomposition of within- and between-person effects

The analytic plan notes that predictors were decomposed into within-person deviations and between-person means. Please elaborate briefly on how this decomposition was performed (e.g., person-mean centering) to aid reader comprehension.

5. Table 1 notation

In Table 1, please clarify in the notes whether “BP SD” refers to between-person standard deviation and “WP SD” refers to within-person standard deviation.

6. Unpleasant feelings and opportunity for compassion

I am curious about the proportion of days on which participants or partners did not report any unpleasant feelings. Because noticing distress is a prerequisite for compassion, it would be informative to examine whether mindfulness predicts the likelihood of perceiving or reporting unpleasant partner emotions, rather than excluding these days entirely.

7. Clarification of mediation results (Hypothesis 2)

In the results section for Hypothesis 2, clarity would be improved by explicitly noting when the total effect (c path) was non-significant but the indirect effect was significant. This distinction is important for correct interpretation of the mediation findings.

8. Moderated mediation

The manuscript reports weak or non-significant evidence for moderated mediation by interdependent self-construal, but the specific pattern of results is not clearly described. Even in the absence of significant moderation, briefly summarizing the observed trends (or lack thereof) would aid transparency.

9. Exploratory analyses

Several exploratory analyses (e.g., those described on pages 20–21) appear somewhat disconnected from the theoretical framework laid out in the introduction. I recommend either (a) better motivating these analyses conceptually earlier in the manuscript, or (b) moving them to the supplementary materials. Given that many of these effects are non-significant or peripheral to the main hypotheses, placing them in the supplement may improve narrative focus.

Reviewer #3 (Remarks to the Author):

Overview:

This study investigated the relationship between state mindfulness and self- and partner- reported compassion, in a sample of 116 romantic couples, assessed for 15 days using a daily diary design. A major strength is the use of intensive longitudinal methods, allowing for the separation of within and between person effects, as well as the inclusion of partner reports of compassion, given the susceptibility for socially desirable responding. However, several limitations warrant consideration. For one, it was hard to determine the novelty or potential broader interest of the work based on the introduction. Furthermore, although the authors emphasize the generalizability of their findings, the sample is restricted to romantic relationships, and it is unclear the extent results generalize beyond this relational context or what can be said about compassion in general (versus directed at a romantic partner). Additionally, the broader implications of the findings are somewhat limited with one study of 230 individuals in romantic relationships, although rich in its dyadic design. The contribution could be strengthened by adding a causal follow-up test or additionally test generalizability and replication, such as examining compassion dynamics among strangers or coworkers. I also had some questions about the statistical models. I expand on these points as well as some additional minor points below.

Introduction:

- It's a little unclear what is already known empirically about the link between mindfulness and compassion. For instance, what did Berry et al find? The authors mentioned an issue regarding previous findings is the generalizability, but it's unclear what is known or the context it had been tested. Additionally, it might be helpful to know the context of past research, was it also in close relationships? Were they assessing state or trait? Were daily diary methods used? Overall, it was hard to determine the advancement of the current study with limited details on past work.

- Is it compassion toward their partner or compassion in general they are observing? Now having read the full manuscript, and knowing it's restricted to the partner, this point should be further elaborated on in the introduction and discussion. What is known about mindfulness in the context of romantic relationships? Has this been tested with similar constructs to how compassion was measured such as responsiveness or social support?

Methods and Analytic Plan

- Do you think the fact that participants were reporting on themselves and their partner on the same constructs may have influenced behaviors? Perhaps this should be noted as a limitation. For instance, do you see changes over time on people's reports of compassion?

- The analysis plan seemed to be missing details. For instance, how was moderation tested and on which pathways? In the script, and later I noticed in the supplemental materials, it seemed like trait measures were controlled for and interactions being included that were not explicitly explained in the analytic plan or pre-registration. Why were traits controlled for? What are the findings without such controls? In general, every interaction and covariate should be listed in the analytic plan.

- I didn't really understand how mediation was conducted or what a stacked multilevel model was. I looked into it, but thought the authors might give a bit more information on this approach or a citation, as it wasn't clear, or preregistered. I also want to note that the preregistration stated they would run a longitudinal moderated mediation, although I am not sure if that was for trait measures. If it was meant for trait, then it doesn't seem the daily mediation analysis was pre-registered, which is fine, but I think should be noted for transparency.

Results

- Sometimes the authors use terms like "Stronger effect" or weak effect, but it's sometimes unclear in what regard. Was this "stronger effect" statistically stronger? What makes the effect weak, when only confidence intervals not including 0 are reported?

- I think it would be helpful to better label the mediation figures (a, b, c, d) and reference figure and letter in the results write up and even the note under the figure as it was not easy to follow.

Discussion

- The authors states "This study supports the generalizability of previous research findings on the relationship between

mindfulness and compassion into people's daily lives," but it's unclear from the intro what this work has found so it's hard to know what is novel here.

- There wasn't any attention to the fact these were romantic couples and compassion was observed just toward one another. How does this factor into the ability to generalize findings? I think conclusions should be drawn with more attention to this limitation.

- The authors stated, "However, present-moment attention may not vary enough within-person from day to day to noticeably relate to compassion at this level". I wasn't convinced by the statement as it seems there could be a few different things happening that may be worth unpacking. Attention likely fluctuates substantially, but variability may not be adequately captured by once-daily retrospective reports. This may represent a limitation of the measurement approach rather than evidence of low true variability. For instance, experience sampling or event-based reconstruction methods might better capture moment-to-moment fluctuations in attention and reduce recall bias. Participants may be motivated to report high levels of attention toward their partner (since the variable is really directed at attending to their partner). This could be evidenced by a ceiling effect, so I went to look at descriptives; yet the reported mean ($M = 9.23$) is difficult to interpret, as attention and acceptance were assessed separately and appear to have originally been measured on a 1–6 scale. Although the table note indicates that variables were rescaled to a 2–12 range to align with other measures, it would be helpful to clarify whether this involved just doubling values and how this rescaling affects interpretation. Notably, attention does appear higher on average than acceptance, which may reflect ceiling effects or social desirability in reports of attention to one's partner. In any case, I didn't find the conclusion that it simply may not vary enough to be a good explanation here.

Version 1:

Decision Letter:

Dear Ms Waclawek,

Your manuscript titled "State and Trait Mindfulness Predict Greater Compassion within Romantic Relationships through Lower Self-Absorption" has now been seen by our reviewers, whose comments appear below. In light of their advice I am delighted to say that we are happy, in principle, to publish a suitably revised version in Communications Psychology.

We therefore invite you to revise your paper one last time to address the remaining concerns of our reviewers and a list of editorial requests. At the same time we ask that you edit your manuscript to comply with our format requirements and to maximise the accessibility and therefore the impact of your work.

EDITORIAL REQUESTS:

SUBMISSION INFORMATION:

OPEN ACCESS:

* DATA AVAILABILITY:

Link Redacted

Best regards,

Jennifer Bellingtier

Jennifer Bellingtier, PhD
Senior Editor
Communications Psychology

Hannah Hao, PhD
Editorial Board Member
Communications Psychology
orcid.org/0000-0002-3342-9132

REVIEWER EXPERTISE:

mindfulness, relationships, dyads

REVIEWERS' COMMENTS:

Reviewer #1 (Remarks to the Author):

The authors have made important changes to their manuscript in line with reviewer feedback. However, a minor amendment is still outstanding which would strengthen the paper:

- introduction: although you have added further information on the definitions of trait vs. state mindfulness, it would be important to add further research on findings from using state mindfulness as an outcome in previous mindfulness studies.

Reviewer #2 (Remarks to the Author):

Thank you for addressing all my feedback extensively. I have no further comments.

Reviewer #3 (Remarks to the Author):

I found the revised manuscript much improved from their initial submission, and appreciated the authors' detailed response letter. My previous comments have been adequately addressed, and I have no further comments. Overall, I believe the work represents a novel contribution that stands to advance understanding as to how facets of mindfulness are linked to social outcomes in everyday relational contexts.

Point-by-Point Response to Reviewers

Reviewer #1 (Remarks to the Author):

This is a well-conducted research study employing the daily diary method, which will inform thinking in the field. While the paper includes thorough research findings, the below may be helpful to consider in revisions to strengthen the paper:

Author response: Thank you for your feedback. It has helped us substantially to improve the manuscript.

Abstract: It would be important to add further details on the participant demographics in the abstract, such as age, gender, length of relationship, sexual orientation, etc. It would also be helpful to include the statistics of the main findings in the abstract. When describing “greater mindfulness”, be more specific on whether you are referring to trait or state mindfulness here.

Author response: Thank you for your suggestion. We have revised the abstract accordingly and added these details.

Introduction: Good to have introduced compassion and mindfulness, but you need to be much more specific and explain the difference between state and trait mindfulness, the theoretical background of this, and include research supporting their respective effectiveness.

Author response: Thank you for this comment. We have now expanded this section in the Introduction to incorporate these points on page 4:

“Despite the evidence for a relationship between mindfulness and compassion, it is still not clear how these processes unfold in people’s daily lives. First, the majority of mindfulness research thus far has measured mindfulness via trait measurements relying on retrospective self-report (Chems-Maarif et al., 2026). Measuring mindfulness as a trait using retrospective report could be a problem because recall bias may limit ecological validity (Enkema et al., 2020; Shiffman et al., 2008). Additionally, trait mindfulness is understood as indicating how frequently and consistently individuals experience mindful states (Brown et al., 2007; Warren et al., 2023). Mindfulness, however, is often conceptualized as an ongoing process (Bishop et al., 2004), and evidence implies that it fluctuates across time (Brown & Ryan, 2003; Warren et al., 2023) and context (Warren et al., 2023). While trait measurements are informative, measuring mindfulness also as a state rather than just an overall tendency allows for a more fine-grained understanding of how mindfulness varies with outcomes of interest. For example, work examining the association between mindfulness and well-being outcomes using both state and traditional retrospective trait measurements found that state measurements produced larger effect sizes than the trait measures (Brown & Ryan, 2003; Enkema et al., 2020; Moore et al., 2016).”

We have also added additional explanation in the Methods section on page 11:

“As described below, daily diary measurements were decomposed into within-person and between-person measurements. The within-person component captures state-like fluctuations, whereas the between-person component captures trait-like individual differences in average

daily levels (Curran & Bauer, 2011). For ease, we refer to within-person as state, and between-person as trait. It is important to note, however, that these “trait” measurements differ from the full-length, retrospective trait questionnaires that participants completed at the beginning of the study.”

We have additionally used the terms state and trait more frequently throughout the manuscript. For example, in the Discussion on page 24 we say “At the within-person, or daily state level, we were able to investigate... At the between-person, or trait level, we were able to investigate...”

A reference is missing on recall bias for mindfulness when recorded retrospectively.

Author response: We have now included references for this on page 4:

“Measuring mindfulness as a trait using retrospective report could be a problem because recall bias may limit ecological validity (Enkema et al., 2020; Shiffman et al., 2008).”

Is there any empirical evidence as to why the daily diary method is an appropriate approach to use here, i.e. has this research been completed with other/similar constructs in the past? If so, it would be important to add this here to hypothesize why this may also be the case for mindfulness and compassion constructs.

Author response: Thank you for raising this point. We have now added references to past research on pages 4–5:

“Daily diary methods have been successfully employed in research on both mindfulness (Donald et al., 2016; Goldberg et al., 2020; Iida & Shapiro, 2017; Perelman et al., 2022) and compassion (Ferreira et al., 2020; Li et al., 2020; Riley et al., 2022)...”

Do not include analysis plans in the introduction section, this needs to be in the methods section.

Author response: Thank you for noting this. We removed this material from the Introduction and now describe it in the Methods on page 16:

“To test self-construal as a moderator of the relationship between self-absorption and compassion, we included a self-absorption by self-construal interaction term.”

Methods: Participants and procedures are generally well outlined in the methods including relevant detail. When adjusting the response scales, was any psychometric analysis completed to account for difference in scoring of these scales? Was factor or Rasch analysis completed to ensure scale validity and/or differential item functioning with the new Likert scoring? If not, it would be important to complete these psychometric analyses to ensure internal consistency of the different scales. This is reported very briefly but needs to be made much more explicit in the manuscript, tables can be added in supplementary materials.

Author response: Thank you for this point. We agree that the manuscript needed greater clarity on this issue. To clarify any potential misunderstanding, we now explicitly state on page 9 that the full-length, retrospective trait questionnaires were not changed from their published versions. Since the daily diary surveys had only two items per construct, we did not complete a

factor analysis, but we do present the correlations with the trait questionnaires (whose response scales were unaltered).

“The daily diary questions used were adapted from published scales to fit the daily diary as well as the dyadic structure of the study; the full-length, retrospective trait questionnaires were unchanged from the original published instruments.”

Results: Generally, well reported, and interesting results!

Author response: Thank you for this positive feedback!

Consider including leading zeros throughout.

Author response: We have now added leading zeros throughout, which improves readability.

It would have been interesting to consider whether relationship length or age influenced these findings, consider repeating analyses where age and relationship length is controlled for.

Author response: Thank you for this suggestion. We conducted these analyses and report the results in the supplementary information (Table S5). We did not observe any change to the pattern of significant results in terms of the main findings reported in the manuscript, except that there was no longer evidence for a within-person interaction between self-absorption and self-construal. Age and relationship length did not themselves predict compassion. Another reviewer suggested we keep exploratory results to the supplementary information to aid in narrative focus, and so we have done this here.

Discussion: Similar to the introduction, more nuance is needed with regards to trait vs. state mindfulness and consolidate your findings with theory and previous research.

Author response: Thank you for this comment. As mentioned above, we now more explicitly incorporate state and trait mindfulness as terms throughout the discussion and have incorporated additional theory and previous research (for example, on page 26).

It would be important to include further limitations of your methods in the limitations section of the discussion, such as limits of your findings with regards to generalisability, i.e. for couples who are part of the LGBTQI+ community, and what else future research needs to focus on in this respect.

Author response: Thank you for pointing out this important limitation. We have now included these considerations in the limitations section on pages 30-31:

“Further, the dyads in our sample were heterosexual romantic couples. This limitation could be addressed by future work investigating if these findings generalize to couples within the LGBTQIA+ community. Beyond romantic couples, future work is also needed to understand the generalizability of these findings to how mindfulness and compassion operate in other interpersonal contexts, for example between strangers, colleagues, or friends.”

Writing in general: there are some typographical errors and incorrect references (e.g. at times, page

numbers are missing for direct quotes), and some references that are very dated. Consider updating this throughout.

Author response: Thank you for pointing this out. We have corrected and updated these issues throughout (for example, we have added the page number for the quote from Gilbert et al. (2017) on page 3 and have added several more recent citations, such as Zhuniq et al. (2025), Lam (2024), Chems-Maarif et al. (2026) and Enkema et al (2020), among others).

Reviewer #2 (Remarks to the Author):

This preregistered daily diary study examines within- and between-person associations among mindfulness, self-absorption, and compassion in romantic couples. Using a dyadic design and partner-reported compassion, the authors provide an important contribution to the literature on mindfulness and social functioning by moving beyond global self-reports and laboratory paradigms to everyday relational contexts. The findings offer a nuanced account of how distinct facets of mindfulness operate differently across levels of analysis and suggest self-absorption as a potential mechanism linking mindfulness to compassion. Below I outline several major and minor points that I believe would strengthen the conceptual clarity, methodological transparency, and interpretability of the findings.

Author response: Thank you for your feedback. Your suggestions have helped has substantially to improve the manuscript.

Major Comments (Conceptual and Analytic)

1. Social desirability and partner reports

The manuscript suggests that partner-reported compassion may mitigate social desirability bias inherent in self-report. While partner reports offer a valuable complementary perspective, I encourage more caution in this claim. Partner reports are also susceptible to social desirability, shared method variance, and relationship-specific biases, particularly in close dyads. It may be more appropriate to frame partner reports as capturing perceived compassion in close relationships, rather than as a cleaner or less biased alternative to self-report.

Author response: We agree and have softened this claim in both the Introduction and Discussion (pages 5-6 and 24):

Introduction:

“Currently, compassion outcomes have been most often studied as compassion towards the self (self-compassion) rather than towards others (Quaglia et al., 2021). To reduce reliance on a single source and to have a better understanding of how mindfulness relates to compassion as it is experienced interpersonally, other-ratings of compassion are needed. Close, important relationships with frequent contact are especially important in dealing with stressors (Feeney & Collins, 2015), making romantic couples who live together an ideal sample. Although ratings from romantic partners are not free from bias, they provide an important interpersonal perspective on compassion.”

Discussion:

“Further, we used partner ratings of compassion to enhance both the validity and ecological applicability of these findings by reducing [reliance on single source self-report and integrating an interpersonal perspective] (this replaced) [potential biases such as social desirability and common source bias].”

2. High overlap between mindfulness attention and self-absorption at the between-person level

The between-person correlation between mindful attention and self-absorption ($r \approx -.70$) is strikingly large. This raises concerns about construct overlap and the interpretability of the between-person mediation model. It would be helpful for the authors to discuss whether these constructs are conceptually and empirically distinct at the trait level, or whether they may reflect opposite poles of a shared underlying dimension. Additional justification (or sensitivity analyses) would strengthen confidence in the mediation findings at the between-person level.

Author response: Thank you for this feedback. We have added additional justification on this important question on page 28:

“In terms of the relationship between mindful attention and self-absorption at the between-person level, we found a strong correlation, which may raise concerns about construct distinctness. However, mindful attention and self-absorption represent dissociable constructs theoretically. For instance, it is possible to be low on mindful attention but to not be focused on the self (for example, one could be distracted by thinking about an interesting topic). The dissociability of these constructs is further supported by the smaller correlation at the within-person level. By contrast, if mindful attention and self-absorption reflected different ends of one dimension, we would expect to find a similarly large correlation at both levels, but this was not supported by the evidence.”

3. Mindfulness and perception of partner compassion

An interesting alternative pathway not addressed in the manuscript is whether individuals higher in mindfulness perceive their partners as more compassionate, independent of their own compassionate behavior. Mindfulness may enhance sensitivity to social cues or positive reinterpretation of partner behavior. Examining associations between one’s own mindfulness and one’s ratings of partner compassion could provide useful context and help disentangle behavioral effects from perceptual ones.

Author response: Thank you for this thoughtful suggestion. Because the APIM includes each member’s mindfulness in the model, this pathway is already represented in our analyses. For example, in the models (Table S2a) with partner-rated compassion as the outcome, the partner mindfulness predictors represent this relationship you mention (the pattern of results for partner mindfulness predicting partner compassion is similar to the results for actor mindfulness). The possibility this represents for future research is now explicitly discussed on page 28:

“Finally, future research should dive deeper into the relational dynamics of mindfulness, self-absorption, and compassion. In our statistical analyses, we used the APIM to properly account for the dyadic structure of the data. However, we did not intend to investigate partner effects as part of our research question. Future research can utilize the partner effects found in this study (reported in the supplementary information) to aid in formulating novel hypotheses concerning these dynamics. For example, when partner-rated compassion was the outcome, partner

mindfulness was also controlled for due to the structure of the APIM. This relationship between partner mindfulness and partner compassion ratings, however, could also be used in formulating future hypotheses about how mindfulness relates to perceptions of compassion.”

4. Divergent within- and between-person effects

The reversal of effects across levels of analysis—where mindful acceptance predicts compassion within persons, but mindful attention predicts compassion between persons—is one of the most intriguing findings of the paper. However, this pattern is not discussed in depth. Greater theoretical elaboration is needed to explain why these facets of mindfulness operate differently across levels. For example, acceptance may be more relevant for momentary interpersonal responsiveness, whereas attention may reflect a stable interpersonal orientation.

Author response: Thank you for this very helpful comment. We have now expanded the discussion and incorporate these thoughts on pages 26-27:

“As noted, day-to-day variability in mindful acceptance was associated with day-to-day variability in compassion. This suggests that an individual’s acceptance fluctuates at a daily level in ways that meaningfully relate to outcomes that are important to people’s lives and relationships. However, mindful acceptance had a weaker relationship at the between-person level. This may signify that it is not acceptance in general that is important, but acceptance at appropriate times (i.e., those in which compassion is relevant). Conversely, the evidence supporting an association between between-person attention and compassion suggests that a general tendency to attend to the present moment may be associated with the capacity to regularly notice environmental cues that elicit compassion, resulting in stronger perceived compassion overall. However, present-moment attention may not vary enough within-person from one day to another to noticeably relate to compassion at the daily level, or in other words, mindful attention may reflect a more stable orientation. We investigated the possibility of a ceiling effect but did not find evidence for this (see supplementary information). It may also be that, when one is not mindfully attentive, the content of their attention becomes relevant. Previous research on mind wandering and caring for others found within-person associations such that mind wandering to something negative or neutral predicted less caring for others behaviour, but not when the mind wandered to something positive (Jazaieri et al., 2016). In line with this result, we did find an indirect relationship between mindful attention and compassion through self-absorption at the within-person level. This evidence may suggest that daily fluctuations in attention are related to compassion when associated with certain other kinds of cognition. This issue is discussed further below. A further important note is that it is possible that fluctuations in attention throughout the day (rather than day to day) may still directly relate to compassion, and more frequent assessment throughout the day may uncover such associations. Further research is needed to test this assumption.”

Relatedly, the compassion measure includes multiple items that may map differentially onto attention versus acceptance. Exploratory analyses examining item-level or subscale-level associations could help clarify whether specific components of compassion are driving the observed effects.

Author response: Thank you for this suggestion. We have run these analyses and report the results in the supplementary information (Table S9). These analyses did not suggest that item-

level differences explain the reversal at the different levels of analysis. However, we discuss the implications of these findings for future research on page 31:

“Additionally, future research may choose to differentiate between compassionate noticing and compassionate acting, as exploratory analyses revealed that mindfulness may be more relevant to others’ perceptions of compassionate acting as compared with compassionate noticing.”

Minor Comments

1. Relationship demographics

Please clarify the relationship-related inclusion criteria (e.g., relationship length, cohabitation status) and consider reporting additional descriptive information about the couples.

Author response: Thank you. We have now added additional information on page 9:

“We recruited a sample of participants from the United Kingdom via Prolific Academic. We invited participants who were over the age of 18, fluent in English, were willing to download an app, had at least an 80 percent approval rate, and were in a heterosexual, romantic relationship with a partner with whom they were cohabiting and who was also willing to participate in the study. No inclusion criteria were specified regarding relationship length. Our total sample comprised 244 participants. Participants were between the ages of 20 and 79, with an average age of 40.33 years ($SD = 11.93$). They were 71% white, 9% Asian, 9% Black, 2% mixed, 1% other, and 8% not reported. Participants were 50% female, 49% male, and 1% not reported; sex was determined through self-reported demographic data provided by Prolific. Sixteen percent of participants reported currently practicing meditation. The dyads had a mean relationship length of 15.01 years ($SD = 10.76$) and a mean cohabitation length of 13.3 years ($SD = 10.62$).”

2. Mindfulness practice

If available, it would be useful to report whether participants engaged in regular mindfulness-related practices (e.g., meditation, prayer), or to note explicitly if this information was not collected.

Author response: Thank you for suggesting this. We now report this information on page 9. This is included in the quote above.

3. Response scale discrepancy

The daily diary response scale is described as ranging from 1–6, yet Table 1 reports means between 8–9. Please clarify this discrepancy.

Author response: This information is in the table note on page 17, but we have added additional information for clarity:

“Mindfulness (attention) and (acceptance) rescaled from 1-6 for this table by doubling values to match the scales of the other variables (2-12).”

4. Decomposition of within- and between-person effects

The analytic plan notes that predictors were decomposed into within-person deviations and between-

person means. Please elaborate briefly on how this decomposition was performed (e.g., person-mean centering) to aid reader comprehension.

Author response: We have now clarified that this occurred via person-mean centering on page 15.

5. Table 1 notation

In Table 1, please clarify in the notes whether “BP SD” refers to between-person standard deviation and “WP SD” refers to within-person standard deviation.

Author response: We have clarified this in the table note on page 17:

““BP SD” indicates between-person standard deviation, “WP SD” indicates within-person standard deviation.”

6. Unpleasant feelings and opportunity for compassion

I am curious about the proportion of days on which participants or partners did not report any unpleasant feelings. Because noticing distress is a prerequisite for compassion, it would be informative to examine whether mindfulness predicts the likelihood of perceiving or reporting unpleasant partner emotions, rather than excluding these days entirely.

Author response: Thank you for this suggestion. We conducted these analyses and report the results in the supplementary information (Table S8). We discuss these findings as a future research direction on page 31:

“Exploratory analyses (reported in the supplementary information) also revealed further possible research directions and important considerations. For example, mindfulness predicted being more likely to report that no unpleasant feelings were experienced on a specific day, both for oneself as well as for one’s partner. Mindfulness therefore may be associated with the dynamics of how these interactions develop and how often compassion is needed or considered an appropriate response in the first place.”

7. Clarification of mediation results (Hypothesis 2)

In the results section for Hypothesis 2, clarity would be improved by explicitly noting when the total effect (c path) was non-significant but the indirect effect was significant. This distinction is important for correct interpretation of the mediation findings.

Author response: Thank you for this suggestion. We have now added this on page 20:

“Although the total effect of attention on compassion was non-significant at the within-person level (see Hypothesis 1), we found evidence for indirect effects via self-absorption.”

8. Moderated mediation

The manuscript reports weak or non-significant evidence for moderated mediation by interdependent self-construal, but the specific pattern of results is not clearly described. Even in the absence of significant moderation, briefly summarizing the observed trends (or lack thereof) would aid transparency.

Author response: We have clarified this point on page 23 by noting that we did not proceed with the moderated mediation analysis in the absence of a significant moderation for our primary outcome of interest:

“We did not find evidence to support the hypothesis that interdependent self-construal moderated the mediation of the relationship between mindfulness and compassion by self-absorption. We began by testing whether self-construal moderated the relationship between self-absorption and compassion. Because we did not find evidence for this moderation for our primary outcome of partner-rated compassion, we did not proceed with the moderated mediation analysis.”

9. Exploratory analyses

Several exploratory analyses (e.g., those described on pages 20–21) appear somewhat disconnected from the theoretical framework laid out in the introduction. I recommend either (a) better motivating these analyses conceptually earlier in the manuscript, or (b) moving them to the supplementary materials. Given that many of these effects are non-significant or peripheral to the main hypotheses, placing them in the supplement may improve narrative focus.

Author response: Thank you for this suggestion. We have moved the exploratory analyses to the supplementary information and agree that this improves narrative focus. We note this on page 24:

“We conducted additional exploratory analyses that are reported in the supplementary information.”

Reviewer #3 (Remarks to the Author):

Overview:

This study investigated the relationship between state mindfulness and self- and partner- reported compassion, in a sample of 116 romantic couples, assessed for 15 days using a daily diary design. A major strength is the use of intensive longitudinal methods, allowing for the separation of within and between person effects, as well as the inclusion of partner reports of compassion, given the susceptibility for socially desirable responding. However, several limitations warrant consideration. For one, it was hard to determine the novelty or potential broader interest of the work based on the introduction. Furthermore, although the authors emphasize the generalizability of their findings, the sample is restricted to romantic relationships, and it is unclear the extent results generalize beyond this relational context or what can be said about compassion in general (versus directed at a romantic partner). Additionally, the broader implications of the findings are somewhat limited with one study of 230 individuals in romantic relationships, although rich in its dyadic design. The contribution could be strengthened by adding a causal follow-up test or additionally test generalizability and replication, such as examining compassion dynamics among strangers or coworkers. I also had some questions about the statistical models. I expand on these points as well as some additional minor points below.

Author response: Thank you for your feedback and advice, which has contributed to substantially improving the manuscript.

Introduction:

- It's a little unclear what is already known empirically about the link between mindfulness and compassion. For instance, what did Berry et al find? The authors mentioned an issue regarding previous findings is the generalizability, but it's unclear what is known or the context it had been tested. Additionally, it might be helpful to know the context of past research, was it also in close relationships? Were they assessing state or trait? Were daily diary methods used? Overall, it was hard to determine the advancement of the current study with limited details on past work.

Author response: Thank you for this suggestion. We revised the Introduction to clarify what was known and how it had been tested. We now also report in more detail on the findings of Berry et al. and added findings from an additional meta-analysis. We address these issues throughout the Introduction (pages 3-6). Some examples:

“In terms of recent empirical work, mindfulness has been reliably linked with compassion. More specifically, recent meta-analyses have found that mindfulness training increases compassion specifically more so than other prosocial behaviours. For example, a meta-analysis by Berry et al. (2020) found that mindfulness training without explicit ethical instruction reliably increased compassionate helping but not instrumental or generous helping. Similarly, a meta-analysis by Kreplin et al. (2018) found that a mindfulness training increased compassion but did not have other prosocial outcomes such as reduced aggression.

Despite the evidence for a relationship between mindfulness and compassion, it is still not clear how these processes unfold in people's daily lives. First, the majority of mindfulness research thus far has measured mindfulness via trait measurements relying on retrospective self-report (Chems-Maarif et al., 2026).”

“Daily diary methods have been successfully employed in research on both mindfulness (Donald et al., 2016; Goldberg et al., 2020; Iida & Shapiro, 2017; Perelman et al., 2022) and compassion (Ferreira et al., 2020; Li et al., 2020; Riley et al., 2022) (though diary studies on compassion so far has been focused on self-compassion, rather than compassion toward others). Research comparing the use of experience sampling methods with traditional retrospective self-report for measuring both mindfulness (Moore et al., 2016) and compassion (Runyan et al., 2019) has shown that experience sampling methods show greater sensitivity and have been able to detect theorized outcomes when retrospective trait measurements have not. Research is needed to bring these strands together to understand the association between mindfulness and compassion in daily life.”

“Currently, compassion outcomes have been most often studied as compassion towards the self (self-compassion) rather than towards others (Quaglia et al., 2021).”

“While there is limited work to date on mindfulness and compassion towards others within romantic relationships, mindfulness has been indirectly positively associated with one's partner's relationship satisfaction through acceptance (Kappen et al., 2018). Mindfulness has also been associated with one's own relationship satisfaction via self-reported caring for one's

partner (Park et al., 2024). Further, studies employing experience sampling methods have found daily mindfulness to positively relate to perceived partner responsiveness as rated by the romantic partner (Perelman et al., 2022), and to self-reported empathy within the context of a romantic relationship (Wen et al., 2022). Research is needed, however, to understand how mindfulness relates to compassion towards one's partner within the context of a romantic relationship - and how it relates to partner perceptions."

- Is it compassion toward their partner or compassion in general they are observing? Now having read the full manuscript, and knowing its restricted to the partner, this point should be further elaborated on in the introduction and discussion. What is known about mindfulness in the context of romantic relationships? Has this been tested with similar constructs to how compassion was measured such as responsiveness or social support?

Author response: Thank you for this comment. We now explicitly state early on that we focus on compassion toward the partner and provide more information on what is known about mindfulness in romantic relationships (pages 5-6). Relevant points from the Introduction are quoted as a response to the point above, and there are additional changes, such as:

"Taken together, to address these previous research gaps and to improve generalizability to daily life, we will use experience sampling methods to investigate how fluctuations in a person's daily mindfulness relate to how compassionate their partner perceives them to be in their usual daily environment. Our first hypothesis is that mindfulness will positively relate to compassion towards one's partner in daily life."

We also mention our focus on compassion toward the partner as a limitation (pages 30-31):

"Further, the dyads in our sample were heterosexual romantic couples. This limitation could be addressed by future work investigating if these findings generalize to couples within the LGBTQIA+ community. Beyond romantic couples, future work is also needed to understand the generalizability of these findings to how mindfulness and compassion operate in other interpersonal contexts, for example between strangers, colleagues, or friends."

Methods and Analytic Plan

- Do you think the fact that participants were reporting on themselves and their partner on the same constructs may have influenced behaviors? Perhaps this should be noted as a limitation. For instance, do you see changes over time on people's reports of compassion?

Author response: Thank you for this interesting thought. We investigated the influence of time and report these results in the supplementary information (Table S10). We discuss the implications of this for future research on pages 31-32:

"Lastly, future research should consider including time as a covariate. Though the focus of this study was not on the dynamics between these variables over time, and including the day of the survey as an exploratory covariate did not change the reported pattern of results, it did reveal that compassion ratings for both self and partner decreased as participants progressed through

the study. It is possible that regularly rating mindfulness and compassion may have affected personal experiences and interactions.”

- The analysis plan seemed to be missing details. For instance, how was moderation tested and on which pathways? In the script, and later I noticed in the supplemental materials, it seemed like trait measures were controlled for and interactions being included that were not explicitly explained in the analytic plan or pre-registration. Why were traits controlled for? What are the findings without such controls? In general, every interaction and covariate should be listed in the analytic plan.

Author response: Thank you for this helpful comment. We have now added details for the moderation analysis on page 16:

“To test self-construal as a moderator of the relationship between self-absorption and compassion, we included a self-absorption by self-construal interaction term.”

Trait level, or between-person variables, represent the decomposition of our daily diary mindfulness measurements into within- and between-person levels to properly conduct our longitudinal model. We now clarify and justify this in the manuscript:

Page 11:

“As described below, daily diary measurements were decomposed into within-person and between-person measurements. The within-person component captures state-like fluctuations, whereas the between-person component captures trait-like individual differences in average daily levels (Curran & Bauer, 2011). For ease, we refer to within-person as state, and between-person as trait. It is important to note, however, that these “trait” measurements differ from the full-length, retrospective trait questionnaires that participants completed at the beginning of the study.”

Page 15:

“Within the daily diary data, to understand the relationships between the constructs of interest both within and between person, we decomposed our predictor variables into between-person means (which we also refer to as trait) and within-person deviations (which we also refer to as state) via person-mean centering.”

Page 16:

“Since our analysis was longitudinal, we included both between-person means (trait level) and within-person deviations (state level) in all models, in order to mitigate possible biasing of the estimates of state-level coefficients with the between-person associations among variables (Curran & Bauer, 2011).”

Regarding interactions, these were included in order to investigate the moderation by self-construal.

- I didn't really understand how mediation was conducted or what a stacked multilevel model was. I looked into it, but thought the authors might give a bit more information on this approach or a citation, as it wasn't clear, or preregistered. I also want to note that the preregistration stated they would run a

longitudinal moderated meditation, although I am not sure if that was for trait measures. If it was meant for trait, then it doesn't seem the daily mediation analysis was pre-registered, which is fine, but I think should be noted for transparency.

Author response: Thank you for this comment. A "stacked multilevel model" describes a way of arranging data with a multilevel structure where the response variable is multivariate, but the model is designed to handle univariate data. Stacking allows modelling 'doubly multivariate' data, 'doubly' because 1) the design has a multivariate structure similar to that of a repeated measures design with a response variable observed at many points in time, and 2) requires modelling more than one variable as a response. In mediation analysis, both the outcome variable and the mediator need to be modelled as response variables. The 'nlme' function in the 'nlme' package allows the specification of a model that treats the response variable and the mediator as a bivariate response vector with a bivariate variance matrix. Thus the 'stacked multilevel model' refers to a technique used to implement the analysis of the preregistered model and doesn't constitute the use of a different model. To clarify this in the text, we have added on page 16:

"To test self-absorption as a mediator underlying the relationship between mindfulness and compassion, we constructed a stacked multilevel model. In this approach, the mediator and outcome variables are stacked into a single response structure, allowing the mediator and outcome equations to be estimated simultaneously within one longitudinal mixed-effects model (Bauer et al., 2006)."

Results

- Sometimes the authors use terms like "Stronger effect" or weak effect, but its sometimes unclear in what regard. Was this "stronger effect" statistically stronger? What makes the effect weak, when only confidence intervals not including 0 are reported?

Author response: Thank you for pointing this out. We have removed any instances in which only confidence intervals not including 0 are reported and have adjusted the language.

- I think it would be helpful to better label the mediation figures (a, b, c ,d) and reference figure and letter in the results write up and even the note under the figure as it was not easy to follow.

Author response: We have updated this throughout pages 20-23 and agree that it improves readability. We now label each mediation a-d and reference the specific mediation figure in the text. We also further explain the figure in the figure note. For example:

"The direct effect of acceptance on partner-rated compassion was non-significant at the within-person level ($c' = 0.07$, 95% BCa CI[-0.05, 0.20]). We found evidence for an indirect effect of acceptance on partner-rated compassion via self-absorption at the within-person level ($ab = 0.06$, 95% BCa CI[0.03, 0.08]). See Figure 1a. The direct effect of acceptance on self-rated compassion was significant at the within-person level ($c' = 0.21$, 95% BCa CI[0.08, 0.32]). We found evidence for an indirect effect of acceptance on self-rated compassion via self-absorption at the within-person level ($ab = 0.05$, 95% BCa CI[0.03, 0.08]). See Figure 1b."

“Note... This figure shows the mediation of the relationship between mindfulness and compassion by self-absorption at the within-person level. The predictors are either mindful acceptance (1a and 1b) or mindful attention (1c and 1d). The outcomes are either partner-rated compassion (1a and 1c) or self-rated compassion (1b and 1d).”

Discussion

- The authors states “This study supports the generalizability of previous research findings on the relationship between mindfulness and compassion into people’s daily lives,” but it’s unclear from the intro what this work has found so its hard to know what is novel here.

Author response: Thank you for raising this concern. We revised the Introduction to clarify what was known and what the contribution of the present study is, which is to investigate the relationship of mindfulness and compassion in daily life via the use of daily diary methods within the context of a romantic relationship. We further contributed by investigating possible mechanisms and moderators.

- There wasn’t any attention to the fact these were romantic couples and compassion was observed just toward one another. How does this factor into the ability to generalize findings? I think conclusions should be drawn with more attention to this limitation.

Author response: Thank you for highlighting this limitation. We now address it explicitly in the Limitations section on pages 30-31:

“Further, the dyads in our sample were heterosexual romantic couples. This limitation could be addressed by future work investigating if these findings generalize to couples within the LGBTQIA+ community. Beyond romantic couples, future work is also needed to understand the generalizability of these findings to how mindfulness and compassion operate in other interpersonal contexts, for example between strangers, colleagues, or friends.”

- The authors stated, “However, present-moment attention may not vary enough within-person from day to day to noticeably relate to compassion at this level”. I wasn’t convinced by the statement as it seems there could be a few different things happening that may be worth unpacking. Attention likely fluctuates substantially, but variability may not be adequately captured by once-daily retrospective reports. This may represent a limitation of the measurement approach rather than evidence of low true variability. For instance, experience sampling or event-based reconstruction methods might better capture moment-to-moment fluctuations in attention and reduce recall bias. Participants may be motivated to report high levels of attention toward their partner (since the variable is really directed at attending to their partner). This could be evidenced by a ceiling effect, so I went to look at descriptives; yet the reported mean ($M = 9.23$) is difficult to interpret, as attention and acceptance were assessed separately and appear to have originally been measured on a 1–6 scale. Although the table note indicates that variables were rescaled to a 2–12 range to align with other measures, it would be helpful to clarify whether this involved just doubling values and how this rescaling affects interpretation. Notably, attention does appear higher on average than acceptance, which may reflect ceiling effects or social desirability in reports of attention to one’s partner. In any case, I didn’t find the conclusion that it simply may not vary enough to be a good explanation here.

Author response: Thank you for this helpful comment. We agree and have expanded this section and include these possibilities among others, as well as avenues for future research on pages 26-27:

“Conversely, the evidence supporting an association between between-person attention and compassion suggests that a general tendency to attend to the present moment may be associated with the capacity to regularly notice environmental cues that elicit compassion, resulting in stronger perceived compassion overall. However, present-moment attention may not vary enough within-person from one day to another to noticeably relate to compassion at the daily level, or in other words, mindful attention may reflect a more stable orientation. We investigated the possibility of a ceiling effect but did not find evidence for this (see supplementary information). It may also be that, when one is not mindfully attentive, the content of their attention becomes relevant. Previous research on mind wandering and caring for others found within-person associations such that mind wandering to something negative or neutral predicted less caring for others behaviour, but not when the mind wandered to something positive (Jazaieri et al., 2016). In line with this result, we did find an indirect relationship between mindful attention and compassion through self-absorption at the within-person level. This evidence may suggest that daily fluctuations in attention are related to compassion when associated with certain other kinds of cognition. This issue is discussed further below. A further important note is that it is possible that fluctuations in attention throughout the day (rather than day to day) may still directly relate to compassion, and more frequent assessment throughout the day may uncover such associations. Further research is needed to test this assumption.”

We also added a further explanation in the figure note:

“Mindfulness (attention) and (acceptance) rescaled from 1-6 for this table by doubling values to match the scales of the other variables (2-12).”

Point-by-Point Response to Reviewers

Reviewer #1 (Remarks to the Author):

The authors have made important changes to their manuscript in line with reviewer feedback. However, a minor amendment is still outstanding which would strengthen the paper:

- introduction: although you have added further information on the definitions of trait vs. state mindfulness, it would be important to add further research on findings from using state mindfulness as an outcome in previous mindfulness studies.

Author response: Thank you for your time and for your helpful feedback on the manuscript. We have now added further research on this topic on page 4:

“State mindfulness itself has been associated with well-being outcomes^{19,24–26}, and work examining associations of mindfulness using both state and traditional retrospective trait measurements found that state measurements were associated with larger effect sizes than the traditional retrospective trait measures when predicting well-being outcomes ^{19,24,25}.”

Reviewer #2 (Remarks to the Author):

Thank you for addressing all my feedback extensively. I have no further comments.

Author response: Thank you for your time and for your helpful feedback on the manuscript.

Reviewer #3 (Remarks to the Author):

I found the revised manuscript much improved from their initial submission, and appreciated the authors' detailed response letter. My previous comments have been adequately addressed, and I have no further comments. Overall, I believe the work represents a novel contribution that stands to advance understanding as to how facets of mindfulness are linked to social outcomes in everyday relational contexts.

Author response: Thank you for your time and for your helpful feedback on the manuscript.